# Authorizing Third-Party Applications Served through Messaging Platforms

**DOI:** 10.3390/s21175716

**Published:** 2021-08-25

**Authors:** Jorge Sancho, José García, Álvaro Alesanco

**Affiliations:** Aragón Institute of Engineering Research (I3A), University of Zaragoza, 50009 Zaragoza, Spain; jogarmo@unizar.es (J.G.); alesanco@unizar.es (Á.A.)

**Keywords:** access control, authorization, messaging platforms, OAuth, virtual assistants

## Abstract

The widespread adoption of smartphones and the new-generation wireless networks have changed the way that people interact among themselves and with their environment. The use of messaging platforms, such as WhatsApp, has become deeply ingrained in peoples’ lives, and many digital services have started to be delivered using these communication channels. In this work, we propose a new OAuth grant type to be used when the interaction between the resource owner and the client takes place through a messaging platform. This new grant type firstly allows the authorization server to be sure that no Man-in-the-Middle risk exists between the resource owner and the client before issuing an access token. Secondly, it allows the authorization server to interact with the resource owner through the same user-agent already being used to interact with the client, i.e., the messaging platform, which is expected to improve the overall user experience of the authorization process. To verify this assumption, we conducted a usability study in which subjects were required to perform the full authorization process using both the standard authorization code grant type (through a web-browser) and the new grant type defined in this work. They have also been required to fill in a small questionnaire including some demographic information and their impressions about both authorization flows. The results suggest that the proposed grant type eases the authorization process in most cases.

## 1. Introduction

The way users interact among themselves and with their environment is constantly changing, and the delivery of digital services continuously evolves to keep pace with these changes. The use of virtual assistants has recently revolutionized the service-delivery paradigm. These assistants interact with users through multiple interfaces (smart speakers, messaging platforms, etc.) providing different kinds of services ranging from simple information enquiries (e.g., the forecast or the air pollution) to more complex operations, such as route optimization or patient follow-up management [1]. As occurs with traditional web applications, virtual assistants might eventually need access to some private information held by a third-party service provider (e.g., the real-time traffic state information) to perform some operations on the user’s behalf. Service providers will require users to authenticate themselves and authorize the virtual assistant to access their remote accounts [2,3,4].

Access to this information is usually provided through Web Application Programing Interfaces (Web-APIs) designed for this purpose [5], while transactions with these kinds of API rely on Hypertext Transfer Protocol (HTTP) for message exchange. Security of Web-APIs usually relies on the Open Authorization (OAuth) 2.0 framework [6,7]. This framework defines a scenario composed of four actors: the end-user, the resource server, the client, and the authorization server. In this scenario, the end-user asks the client to perform some operation on his behalf so that the client requires access to some user’s protected resource stored at the resource server to complete the requested operation. The authorization server arbitrates the access to the protected resource, ensuring that the end-user has sufficient rights to access the requested resource and that the client has been authorized to act on the end-user’s behalf. In this context, OAuth includes all the required elements to empower the end-user, allowing him to authorize the client to act on his behalf, without having to show his credentials. Among other things, it defines a protocol (grant type) that allows the authorization server to directly interact with the resource owner, thus verifying his identity and gathering his consent to let the client act on his behalf. This grant type was designed with a concrete type of client profile in mind, that is, web-applications that are served by a web server and accessed by end-users using a web-browser as user-agent. Thus, this grant type was conveniently designed so that the interaction between the resource owner and the authorization server could be performed through the same interface (user-agent) that the resource owner was already using to access the client, i.e., a web browser.

When communication channels other than web-browsers are used for the interaction between the end-user and the client, such as messaging platforms, the underlying authorization problem is essentially the same (providing the end-user with a reliable method to express his intention of authorizing the client to act on his behalf), but some considerations must be taken into account. These considerations include two aspects: dealing with security issues and usability aspects derived from changes in the scenario. The contributions of this paper are as follows:We propose a new OAuth grant type to be used when the interaction between the client and the resource owner is done through a messaging platform. Unlike the standard OAuth authorization code grant type, it allows the authorization server to ensure that no Man-in-the-Middle (MitM) risk exists between the resource owner and the client before issuing the access token so that no user’s private information is exposed to a potential attacker. This new grant type has been designed to allow the authorization server to interact with the resource owner using the same communication that he was already using with the client, the messaging platform.We conducted a usability test to verify that gathering the user consent using the proposed grant type results in a better overall user experience of the authorization process.

The rest of this paper is organized as follows. In Section 2, we describe the access-control background. Section 3 shows the proposed grant type. Section 4 presents a security analysis, while Section 5 describes the usability study. Section 6 discusses the results obtained. Finally, the conclusions and future work can be found in Section 7.

## 2. Background

### 2.1. OAuth 2.0 Framework

OAuth 2.0 is currently the standard framework for authorization. It defines a base scenario composed of four actors: the end-user, the resource server, the client, and the authorization server. The end-user is an entity capable of granting access to a protected resource that is stored in the resource server. The client is an application that requires access to the protected resource to perform some action on behalf of the end-user and with his consent, while the authorization server arbitrates the access to the protected resource by means of two endpoints: the authorize endpoint and the token endpoint. When the client requires access (on behalf of the end-user) to the protected resource, it sends an authorization request (A) to the authorize endpoint so that the authorization server can directly interact with the end-user, authenticating him and gathering his consent to authorize the client to act on his behalf. If the end-user authorizes the client to act on his behalf, the authorization server sends an authorization code in the authorization response to the client (B). The client exchanges the code obtained for an access token at the token endpoint of the authorization server, sending the code in the token request (C) and receiving the access token in the body of the token response (D). Requests to the token endpoint must include the client credentials whenever client authentication is required. Finally, the client can use the obtained access token to access the protected resource at the resource server. The standard defines a protocol based on HTTP redirections (see Figure 1) so that the authorization server can directly interact with the end-user. In this flow, information is passed from the client to the authorization server encoded as query parameters in the URL when redirecting the end-user.

### 2.2. TextSecure Security

Most messaging platforms, such as Signal, WhatsApp, Facebook Messenger, or Skype, rely on the TextSecure protocol [8] to provide security to users’ conversations. This protocol provides end-to-end encryption between communication peers so that no eavesdropper is able to see the content of exchanged messages. To that end, it relies on a public key scheme where each user-agent (i.e., application installed in the user’s device) generates its own key pair (public and private keys) at the time of installation. This key pair is further used to generate session keys that are used to encrypt the conversation.

However, there is no way for a party to be sure a priori that a given public key belongs to his communication peer. A MitM attacker would be able to modify messages exchanged at the start of a conversation, tricking parties into believing that his own public key (the attacker one) belongs to the other communication peer, developing the attack. This is addressed by the so-called authentication ceremony, which consists of comparing parties’ public keys using an out-of-band channel, thus preventing this risk. In current messaging platforms, it can be done by comparing a safety number (which is really a concatenation of both users’ public key fingerprint) or scanning a QR code.

### 2.3. Problem Definition

In this work, we will focus on a scenario (see Figure 2) in which an end-user is using a messaging platform to access a remote service provided by the client. The end-user requires to authorize (in an OAuth sense) the client to access, on his behalf, some protected resource held by the resource server.

Currently, if the client needs to access a resource stored at the resource server and this access has been secured using the OAuth framework, the client’s best attempt to obtain the user consent consists in sending a link to the user as a normal message in the conversation pointing to the authorize endpoint of the authorization server with the required parameters (the same URL to which the user would be redirected if the communication between the user and client were through a web-browser). The user must follow this link and continue the interaction through the web-browser to authorize the client to access his protected resource. Once the client obtains the required access token, the client will again interact with the user through the messaging platform so that he must return from the web-browser to this platform. Switching from the messaging platform to the web-browser and back may hamper the usability of the system, making users reluctant to use new communication channels to consume services.

Given the construction of the TextSecure protocol, a MitM attacker could be placed between the user and the client, inevitably seeing all the private information being exchanged between both. Even worse, this attacker would be able to modify the link sent by the client (to start the authorization process) so that it will point to a fake website owned by the attacker pretending to be the authorization server. In such a situation, the attacker would be able to trick the user into introducing his private credentials at the fake site, making them available to the attacker. This MitM threat could be simply prevented by the authentication ceremony (as stated in Section 2.2). However, users must complete the authentication ceremony with every single client they want to interact with, which would hamper the user experience. Studies point out that most users do not complete the authentication ceremony even to exchange sensitive information, such as credit cards numbers [9,10,11], and the worst part is that the authorization server, which is the custodian of the users’ data privacy, has no way of ensuring that the authentication ceremony has been completed between them before issuing an access token.

## 3. Proposed Authorization Protocol

In this section, we detail the proposed grant type (see Figure 3). It was designed to be used with client profiles that use messaging platforms as user-agents to deliver services. In this new grant type, the authorization server is able to interact with the end-user using the same interface (user-agent) that he is already using to interact with the client, i.e., the messaging platform. Messaging platforms do not support any kind of redirection mechanism, which is crucial in the Authorization Code grant type defined by the OAuth standard. These redirections have a double function: to allow the client to pass the required information to the authorization server (and the other way) and to allow the authorization server to directly interact with the end-user (to ask them for their consent). As can be seen in Figure 3, our protocol substitutes those redirections with direct HTTP communication between the client and the authorization server (to pass the required information), while the direct communication between the authorization server and the end-user is enabled thanks to the inclusion of two new parameters (platform and id) at the authorization request as explained below.

The proposed protocol also integrates the security-related aspects as a part of the protocol. The authorization server is also allowed to complete the authorization process securely without relying on the authentication ceremony between the user and the client. It is also able to guarantee that no MitM risk exists before letting the client obtain any private information.

### 3.1. Prerequisites

Since the resource owner will interact with the authorization server using a messaging platform, some prerequisites are needed before the proposed grant type can be used. There are two main prerequisites: obtaining the user identifier (id) on a specific platform and completing the authentication ceremony on this platform (as shown in Section 2.2). This must be done using an out-of-band channel and would typically be performed at the time the user registers himself at the authorization server using its web interface. Note that an authorization server may support several messaging platforms. In that case, the user would firstly be asked to select which platform is going to be registered. On the other hand, the same user may register himself on several platforms.

Most messaging platforms currently support two ways of completing the authentication ceremony: by comparing a safety number or by scanning a QR code. We present three different ways of doing this. The first way is asking the user to manually write the safety number in a text box. The second way, which is appropriate when the registration is being done from a different device from that where the messaging application is installed (i.e., a laptop), is asking the user to show the pairing QR code to the webcam of the device where the registration is taking place. The third way, which is appropriate when the registration is being done from the same device that has the messaging application installed (i.e., a smartphone), is asking the user to take a snapshot of the pairing QR code and upload it to the authorization server.

### 3.2. Authorization Request

When the client application requires access to a protected resource, it sends a request to the authorize endpoint of the authorization server, including the following parameters using the “application/x-www-form-urlencoded” format with a character encoding of UTF-8 in the HTTP request entity body:

 

platform

REQUIRED. The concrete messaging platform the user is interacting with (e.g., WhatsApp, Signal).

 

id

REQUIRED. The user identifier in the messaging platform (e.g., phone number).

 

key_fingerprint

REQUIRED. Fingerprint of the long-term identity public key.

 

client_id

REQUIRED. A string uniquely identifying the client.

 

redirect_uri

OPTIONAL. A URI to redirect the user back after the authorization.

 

scope

OPTIONAL. A string describing the access rights requested by the client.

 

state

RECOMMENDED. An opaque value used by the client to maintain state between the request and callback. 

 

Among these parameters, there are some defined in the standard (*client_id*, *redirect_uri*, *scope,* and *state*) and others defined specifically for this grant type (*platform*, *id,* and *key_fingerprint*). The *platform* parameter is used to determine which specific messaging platform the resource owner is using to interact with the client (e.g., WhatsApp, Signal), and the *id* is the identifier of the resource owner at this platform (e.g., the user’s phone number). These parameters are required to allow the authorization server to contact the resource owner since his identity is a priori unknown. The *key_fingerprint* is the fingerprint of what the client believes to be the end-user’s public key. It is included to allow the authorization server to ensure that no MitM attack between the client and the end-user is taking place before issuing the access token even if the authentication ceremony between them has not been completed.

### 3.3. Authorization Request Processing

When the authorization server receives this request, it is validated, ensuring that all the required parameters are present and valid. Then, the authorization server verifies that the received combination of *platform* and *id* has previously been registered for any user. If so, the *key_fingerprint* received from the client is verified to ensure that it is the same as that shown to the authorization server (the fingerprint of what is currently being shown to the authorization server as the end-user’s public key). They are also compared to be the same as that stored as a result of the authentication ceremony performed as explained in Section 3.1 (the fingerprint of the actual end-user’s public key). If everything goes as expected, the authorization server acknowledges the received request, sending a 200 OK response to the client. Otherwise, the authorization server quickly rejects the authorization, sending a 400 Bad Requests message, including the error response parameters (*error*, *error_description*, *error_uri,* and *state*) in the response body as defined in the standard. If the provided combination of *id* and *platform* has not been registered previously by any user, the error parameter is set to “access_denied”. If the public key fingerprint sent by the client does not match the one registered in the authorization server for that combination of *id* and *platform*, the authorization server returns the error parameter set as “public_key_not_match”. It can be used by the client to inform the resource owner about a possible MitM attack and abort the communication when considered necessary.

At this point, the authorization server can already contact the resource owner at the platform and id specified by the client. In this interaction, the authorization server may ask the user for some extra authentication information, like a one-time password (OTP) or some voice biometrics, or simply rely on the possession of the device where the messaging application is running (i.e., the possession of the complementary private key of the public key that was associated to a specific user during the authentication ceremony at the time of registration). Once the user identity has been verified, the authorization server evaluates the access control policies as it normally does (the specific policy evaluation method lies outside the scope of the OAuth standard). If everything goes as expected, the authorization server contacts the resource owner again to inform him the client application is requesting access to a resource on his behalf and asks him to authorize the client to complete this operation.

### 3.4. Authorization Response

Once the authorization server has obtained the resource owner’s consent, it sends the authorization response to the client endpoint specified at the time of client registration or in the authorization request by the *redirect_uri* parameter. This response includes the following parameters in the HTTP request entity-body using the “application/x-www-form-urlencoded” format with a character encoding of UTF-8.

 

code

   REQUIRED. The authorization code generated by the authorization server.

 

state

   REQUIRED if the “state” parameter was included in the client authorization request. The exact value received from the client.

 

If the resource owner’s consent cannot be obtained, the parameters included in the response are those defined in the error response (*error*, *error_description*, *error_uri,* and *state*), setting the error parameter as “access_denied”. Independently of the result of the authorization and the parameters included in the authorization response, the client acknowledges the reception of the authorization response, sending a 200 OK response.

### 3.5. Obtaining Access Token

Finally, if the client has obtained a valid authorization code, it is exchanged for an access token in the token endpoint of the authorization server using the token request and token response defined in the standard, authenticating the client when required. Once the client application has obtained the access token, it can obtain the required resource from the resource server, authorizing the operation with the access token.

## 4. Security Analysis

In this section, we analyze the security provided by the proposed authorization flow. First, we show how this method allows the authorization server to prevent the existence of a MitM between the end-user and any client, only requiring that the authentication ceremony between the end-user and the authorization server has previously been completed. Secondly, we analyze the degree of security of the proposed method in the face of known attacks against OAuth. Other threat models (e.g., Denial of Service or eavesdropping) have not been considered in this section, as they are out of the scope of the manuscript.

### 4.1. MitM Attacker

All the trust in the TextSecure protocol is based on asymmetric cryptography. Each participant generates its own key pair (a public key, $K_{x}^{+}$, and a private key, $K_{x}^{-}$) that is used in the generation of all the subsequent cryptographic material required to encrypt and sign all messages exchanged during the conversation. Thus, a MitM attacker has to cheat both communication ends to make one communication end think that the attacker’s public key ($K_{e}^{+}$) actually belongs to the other communication end [12]. In the scenario shown in Figure 4, the user has been cheated into thinking that the attacker’s public key ($K_{e}^{+}$) really belongs to the client. In the same way, the client has been cheated into thinking that the attacker’s public key ($K_{e}^{+}$) really belongs to the user. Finally, the user can be sure that what he believes to be the authorization server’s public key ($K_{as}^{+}$) belongs to the authorization server and, in turn, the authorization server can be sure that what it believes to be the user’s public key ($K_{u}^{+}$) really belongs to the user thanks to having completed the authentication ceremony (which is a prerequisite of the proposed protocol).

In such a scenario, the MitM would be able to see and modify messages exchanged between the end-user and the client, while the authorization server has no way of detecting his presence if no additional measures are applied. The use of the key_fingerprint parameter in the authorization request of the proposed protocol is intended to sort out this situation. When the client starts the authorization process, the fingerprint of the attacker’s public key ($K_{e}^{+}$) is included in the request, since the client has been cheated into thinking that this public key actually belongs to the end-user. When the authorization server receives the authorization request, it is able to compare the fingerprint of the received public key ($K_{e}^{+}$) with the fingerprint of the public key that it has stored for the end-user ($K_{u}^{+}$) as a result of the authentication ceremony. Any time that a MitM appears between the end-user and the client, the public key fingerprints will not match, and the authorization server would be able to detect its presence before issuing any access token.

### 4.2. Security against Known Attacks

In this section, we compare the security provided by the standard authorization code grant type through a web browser as suggested in Section 2.3 (method A hereafter) with the new grant type proposed in Section 3 (method B hereafter). To that end, we analyze how attacks described in [13] affect these solutions or not, considering the five attacker models defined in [13] and the MitM attacker presented previously (Section 4.1). We classify these attacks in three groups. The first group includes attacks that do not depend on the grant type flow but are related with other architectural aspects of the OAuth framework. These attacks include Access Token Leakage at the Resource Server, TLS Terminating Reverse Proxies, Refresh Token Protection, and Client Impersonating Resource Owner. Attacks in this group affect both methods as they are independent of the specific grant type, and its countermeasures should always be applied. The second group includes those attacks that are exploited taking advantage of the HTTP redirection mechanism (used in the original OAuth Authorization Code grant type to move the user from the client to the authorization server and back) or any feature related with the web-browser (like the browser history). Thus, they would affect method A but would not affect method B, at least in its current form, as the proposed method does not rely on any HTTP redirection nor the use of a web-browser. This group includes Credential Leakage via Referer Headers, Credential Leakage via Browser History, Authorization Code Injection, Access Token Injection, Cross Site Request Forgery, Open Redirection, and Clickjacking. Finally, attacks in the third group, which are the most interesting for us, affect both methods in different ways and are analyzed more carefully below. These attacks are Insufficient Redirect URI Validation and Mix-Up Attacks. Table 1 summarizes this information.

The Insufficient Redirect URI Validation attack, as described in [13], is conducted as follows. First, the attacker needs to trick the user into opening a tampered URL in his browser that launches a page under the attacker’s control. This URL initiates an authorization request with the client ID of a legitimate client to the authorization endpoint, including a redirect URL under the attacker’s control and matching the registered redirect URL pattern for the legitimate client. The authorization request is processed and presented to the user. If the user does not see the redirect URI or does not recognize the attack, the code is issued and immediately sent to the attacker’s domain. When using method A, the MitM only needs to tamper with any legitimate authorization link sent by the client so that the redirect_uri points to a domain under his control. In this case, it would be difficult for the user to notice that he is being attacked given that starting the authorization by opening the link is part of the legitimate authorization using method A. On the other hand, when using method B, the attacker’s best attempt would be to trick the user into believing that a client under his control is actually a legitimate client (e.g., initiating a new conversation with the user and stating that it is a known client application whose phone number has changed recently) and to obtain the user’s consent to access his protected resources. Independently of the method used, this could be shortcut by strictly validating redirect_uris (i.e., performing strict string matching instead of supporting regular expressions) at authorization server.

Mix-Up attacks require the client to try to obtain authorization from the user using an authorization server under the attacker’s control. When using method A, the MitM can simply modify the user messages to trick the client into thinking that the user has selected the authorization server under the attacker’s control when he actually has not. On the other hand, when using method B, this attack would only be possible if the user intentionally selects the authorization server under the attacker’s control for any reason. This attack could be prevented by the client using distinct redirect URIs for each authorization serv-er.

Finally, there is the passive MitM attack where the MitM is placed between the client and the user without modifying any message, with the sole purpose of eavesdropping on the user’s private information exchanged from his routine use of the client. When using method A, this kind of attack can be prevented by completing the authentication ceremony between each user and each client. However, as already stated, the authorization server, which is responsible for the security of the user’s resources, has no way of being sure that this authentication ceremony has been performed before issuing an access token. Method B prevents this attack as explained in Section 4.1.

## 5. Usability Study

We conducted a study to better understand how users perceive the proposed authorization method and what might make them reluctant to use it. This study consists of two tests: test A and test B. In both tests, the subject is required to interact with a virtual assistant, Alfred, using the Signal secure messaging application [14]. At a certain point of the conversation, Alfred informs the user that he needs his authorization to access some protected resource on his behalf. In test A, subjects are requested to complete the authorization process using the standard authorization code grant type through a web-browser, as suggested in Section 2.3 (see Figure 5a). In test B, they are requested to authorize Alfred using the new grant type proposed in Section 3 (see Figure 5b). After completing both tests, subjects are required to fill in a small questionnaire, including some demographic information and their impressions about both authorization methods.

### 5.1. Study Recruitment, Design and Realization

The study participants were recruited from our campus and from our circle of acquaintances in equal parts. We ensured that none of them previously knew what our work consists of or the objective of the study to avoid biased results.

We designed the study so that each subject would be provided with two similar smartphones (one for each test) with a preregistered virtual assistant contact. The reason behind using different smartphones for each test was that it eases the subjects’ understanding of what they are doing (authorizing the virtual assistant by two different means), as we saw during the study design, providing a more reliable feedback.

When the participants arrived, they were firstly asked to read and sign the informed consent. After that, we briefly explained the basis of the study and informed them that a study coordinator would be observing their interaction and would answer any possible question they might have. At this point, the study coordinator handed out the smartphone prepared for test A and provided the following context information:


*Suppose that you are using the Signal app to normally interact with your virtual assistant, Alfred. You ask him when the following appointment with the physician is. The objective is to complete the required steps to obtain this information from Alfred.*


After successfully completing the first test, the subject was provided with the other smartphone (prepared for test B) and instructed to repeat the task after being warned that some steps in the process would be different. After completing the task, the subject was required to briefly explain to the study coordinator what he/she had done to check the subject’s understanding of the technology (up to a certain point). The subject was then required to complete a small summary containing some demographic questions and some related to their impressions of both tests. The demographic questions include the subject’s gender and age. The subjects also had to select one of three options describing their degree of familiarity with the technology. The three levels were “Occasional user”, “Habitual user”, and “Advanced user”, defined as follows:Occasional user: your main use of computers is to occasionally navigate the web, send/read emails, or see some videos on YouTube.Habitual user: you usually rely on a computer for many tasks daily and/or part of your work depends on it as a user.Advanced user: you are a computer enthusiast and/or your work involves a deep level of computer understanding (programmer, computer sciences, etc.).

For each test, the question “What has been your impression on the usability of method X?” was asked to rate the usability of the authorization method. Possible answers to this question were integers from 1 to 10, where a higher score was better. The question “Do you believe the method X to be secure? Why?” was also included for each test, where possible answers were “yes” and “no” along with a space to justify their answer. Finally, the participants were requested to answer the question “Which method would you prefer to use?” considering their overall experience and taking into account both usability and security. A text box was also required to be filled in, including some “Specific comments that motivate your previous responses”. After completing the questionnaire, the study coordinator announced that the study had finished.

### 5.2. Demographics

A total of 24 participants took part in the study. One of them was a priori excluded from the study given that he affirmed that he did not understand what he had done after completing the test. The remaining 23 participants were categorized in accordance with three parameters, their gender, their age, and their degree of experience of interacting with computers. Ten participants out of the 23 were male (43% of the total). The participants were categorized in three age groups: under 25, between 25 and 48, and 49 and over. The first group had 8 participants (35% of the total), the second group 10 (43%), and the last group 5 (22%). The level of familiarity with computers was categorized in the three levels defined in Section 5.1: “Occasional user”, “Habitual user”, and “Advanced user”. The first group had 10 participants (43%), the second group 4 (17%), while the last group had 9 (39%). All this information is summarized in Table 2.

We can see that the population is reasonably well-balanced as regards the gender of the participants. Looking at their ages, the younger and mid-range groups are also well balanced while the oldest subjects’ group has less members than the others. Finally, the participants experience with the use of computers is skewed since only a few are habitual users. This is due to the recruitment procedure. Most of the participants from our circle of acquaintances are occasional users, while participants from our campus are advanced users. However, the occasional users and advanced users’ groups are well balanced.

## 6. Results and Discussion

In this section, we present the results obtained from the usability study both qualitatively and quantitatively. The participants’ responses to the questionnaire are summarized in Table 3. The first row in the table includes the mean number of points with which participants rated the usability of both methods out of a maximum of 10. The second row shows the number of participants who believe each method to be secure, while the last row shows how many participants preferred one method over the other.

There is no significant difference between the usability rates obtained by both methods. As many participants stated, “both methods are very simple to use”. However, the method proposed in this work obtained a slightly better result. In Table 4, we can observe participants’ comments that justify this. From the usability point-of-view, most participants who preferred method B said that it is simpler because they do not have to leave the application to complete the process (15 participants). A smaller set of participants stated that they prefer method A since the interaction is more similar to the one that they currently use for authorization tasks (just three participants). From the security perspective, 13 out of the 23 participants believed the method used in test A to be secure. Many participants (16 out of 23) expressed their concern that they do not feel comfortable clicking the link provided by the client. However, some of them (six) still considered this method to be secure. On the other hand, all the participants involved in the study believe the method used in test B to be secure. Just one participant pointed out that he obtained a better security impression with the method for the test A stating that “seeing the link makes me more comfortable as I get a deeper understanding about how the system works”. The conjunction of all these facts explains that most participants (19 out of 23) prefer to use the method proposed in this work.

Figure 6 details the results showing differences between the population groups described in Section 5.2. Figure 6a shows results distributed by gender, Figure 6b shows their distribution by ages, while Figure 6c does the same with the level of familiarity with the technology. In all the subfigures, the bars are grouped in three categories: the usability rate, security, and the preferred method. The first group of bars represents the usability rate assigned to each test (over a maximum of 100 points), the second group shows the percentage of participants that believe the method to be secure, while the third group shows the participants’ preferences of one method over the other. The bars for the same demographic group share the same color between categories, while the solid bars represent results for Test A and the hollow bars for Test B.

Figure 6a shows no significant differences in how participants of different genders rated the usability, although male participants believe method A to be more secure than female participants, which is also reflected in the preferred method for each of them. Figure 6b, which shows the results split by the age of the participants, indicates that there are no significant differences among users in the 18 to 24 age group and users in the 25 to 48 age group in any of the three categories. However, older participants (49 and over) rated the usability of method A as being worse than method B and had less confidence in the security of method A. This is reflected in the fact that no user in this group preferred method A over method B.

The most interesting results can be seen in Figure 6c, which shows the results depending on the technical abilities of the participants. In this figure, occasional users are labeled as tech. 1, habitual users as tech. 2, and advanced users as tech. 3. The more experienced participants rated the usability of method A higher than others with less experience. A larger number of experienced participants also trusted method A to be secure compared with participants in other groups. Finally, all the participants that preferred method A over method B were advanced users. Some insights derived by this study may be conditioned by the limited number of participants, and a study addressing this concern would be needed to confirm our findings.

As a final comment, none of the participants involved in the study noticed that the link sent by the client points to a HTTP service (see Figure 5a), which is not using TLS to secure the connection (i.e., using HTTPS instead). In a real scenario, this link might be sent by a MitM (trying to cheat the user) if the authentication ceremony has not been completed between the user and the client. This demonstrates that most users (including some graduates in computer sciences) are far from understanding all the security implications of their decisions and actions. Thus, security methods should be designed to protect users’ security on their behalf, reducing their exposure to possible threats derived from their actions.

In this context, minimizing the number of required authentication ceremonies would improve not only the system usability but also the security of the communication. Using the flow proposed in this work, with the OpenID Connect [15] protocol to provide federated identity, would help with this problem. In the OpenID Connect protocol, the client application wants to obtain some information about the end-user (such as their identity) from an identity provider. One of the OAuth grant types is used to allow the identity provider to authenticate the user and obtain their consent to share the information with the client. A client application that uses the OpenID Connect protocol with the proposed grant type to deal with users’ identities would not need to worry about the presence of a possible MitM attacker even without completing the authentication ceremony. In this case, the identity provider would seamlessly verify that the public key fingerprint included in the authorization request really belongs to the user (just in the same way that an authorization server does as part of a normal authorization flow), ensuring that no MitM risk exists. Thus, the user is only required to complete the authentication ceremony with the identity provider instead of doing so with each client.

In the same way, an authorization server may rely on an identity provider to deal with a user’s identity. In such a situation, the authorization server would act as the client of the identity provider so that only the authentication ceremony with the identity provider would newly be required. This is especially interesting for those scenarios where users’ resources are spread across several resource servers protected by different authorization servers.

## 7. Conclusions

In this work, we propose a new protocol to allow users to authorize third-party applications when the interaction with these applications is taking place through a messaging application. This protocol has been designed as a new OAuth grant type to take advantage of all the elements already defined in this framework and to provide direct access to all existent APIs that are already secured using it. The proposed grant type allows the authorization server to interact with the resource owner directly through the same messaging platform already being used for interaction with the client. It also allows the authorization server to be sure that there is no risk of an MitM between the client and the user before issuing an access token.

Aligning the way that authentication and authorization tasks are handled with how users interact to obtain the service that requires these tasks improves the overall system usability. In the usability test, we saw that most users found the proposed method usable for authorizing clients through messaging platforms and preferred it to using a web-browser with this same purpose. This is especially true for those users less experienced with computers, who rated our proposed approach highly for both usability and security. This is very important since users with less technical skills are the main target of the new service-delivery paradigm aiming to reach more population sectors.

The use of new features included in some messaging application (like Telegram’s embedded web browser) may affect the usability test results, as they make the authorization process smoother without forcing the user to swap between the messaging platform and the system web browser. However, some major messaging applications (i.e., Signal, WhatsApp, etc.) do not support this feature yet, and the problem with the MitM attacker would always remain even with the use of embedded browsers.

Finally, the proposed method widely reduces the number of authentication ceremonies that must be completed. However, one authentication ceremony (with the authorization server) is still required. Further research may explore the inclusion of digital certificates to stop relying on the completion of any authentication ceremony. On the other hand, some insights derived from the usability study may be conditioned by the limited number of participants (a total of 23), and a study addressing this concern would be needed to confirm our findings.

## Figures and Tables

**Figure 1 sensors-21-05716-f001:**
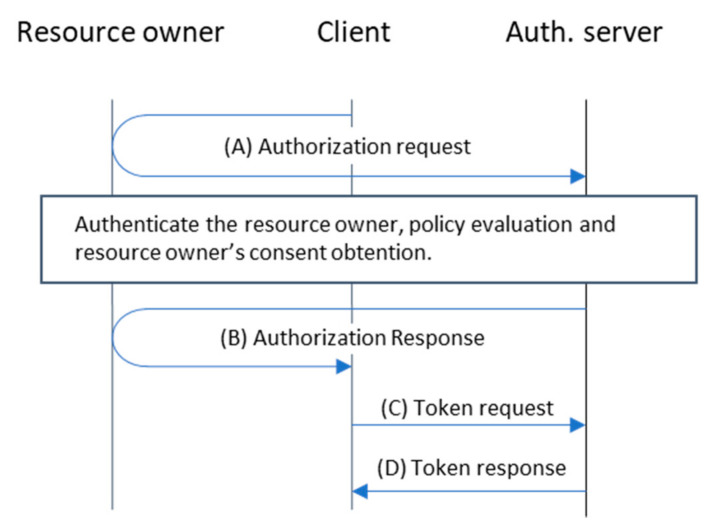
OAuth 2.0 Authorization code grant type.

**Figure 2 sensors-21-05716-f002:**
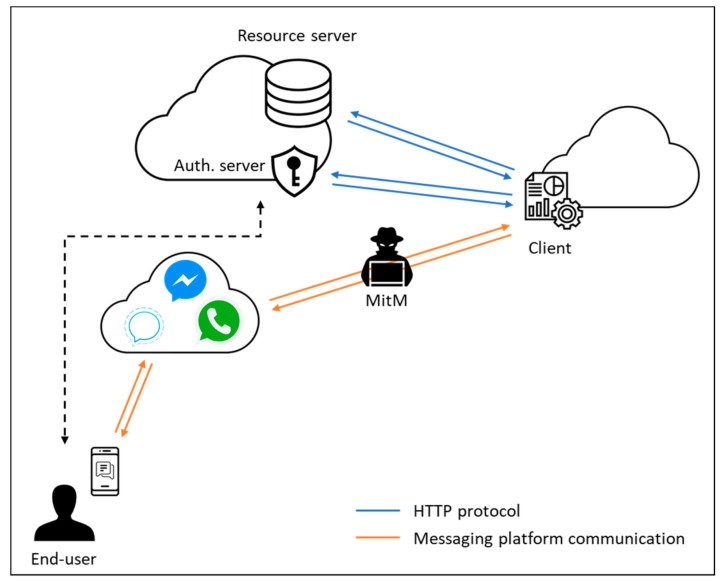
Reference scenario. End-user authorizes the client to access his resources at the resource server on his behalf.

**Figure 3 sensors-21-05716-f003:**
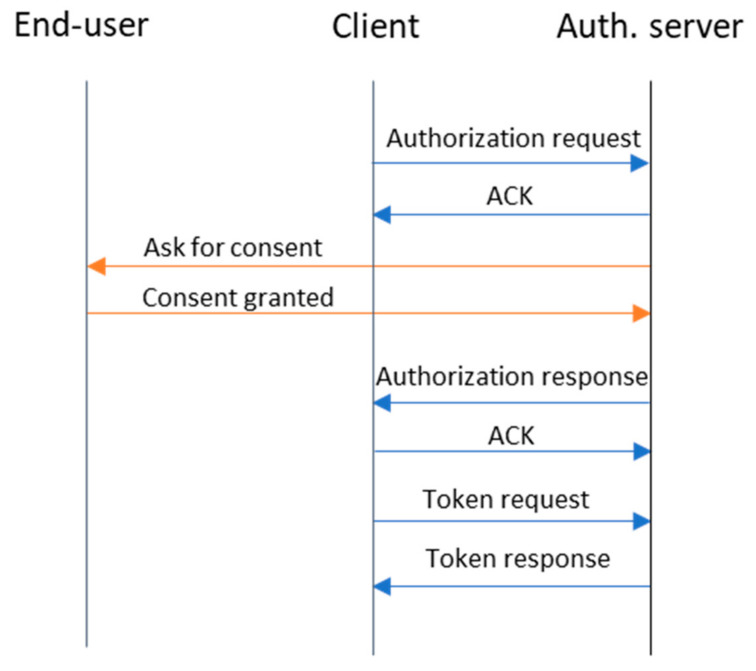
Proposed grant type to authorize third-party applications served through messaging platforms. Blue arrows represent HTTP messages, while orange arrows represent communications through the messaging platform.

**Figure 4 sensors-21-05716-f004:**
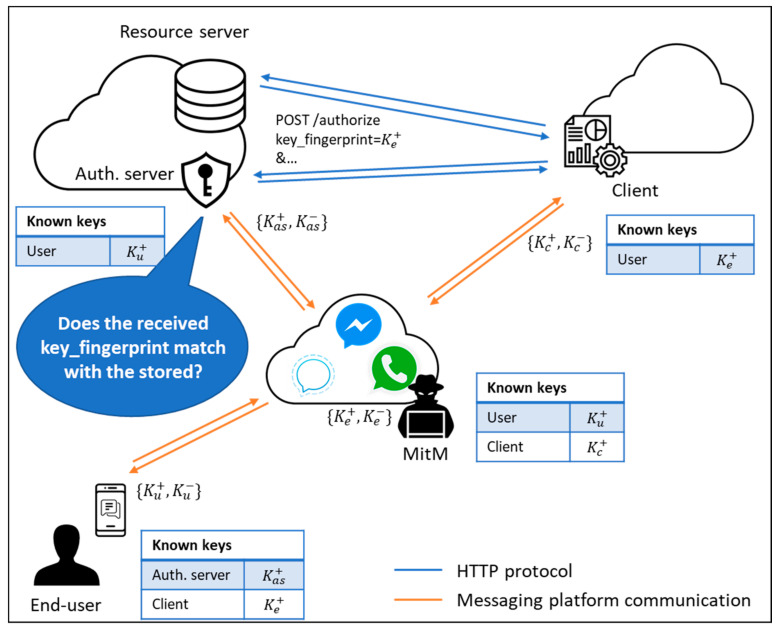
Man-in-the-Middle prevention showcase using the proposed grant type.

**Figure 5 sensors-21-05716-f005:**
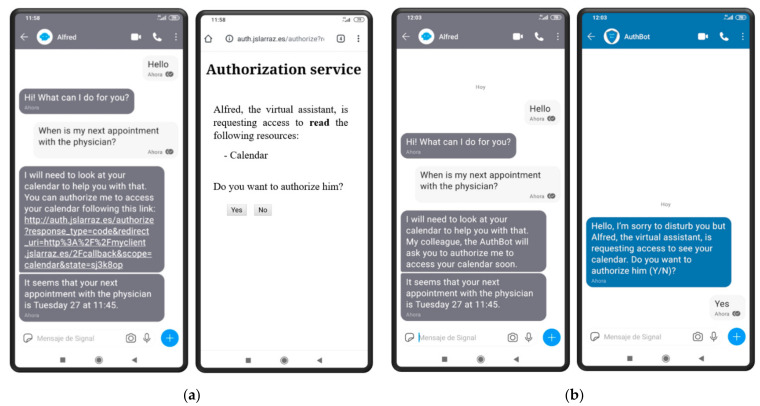
(**a**) Example interaction in test A; (**b**) example interaction in test B.

**Figure 6 sensors-21-05716-f006:**
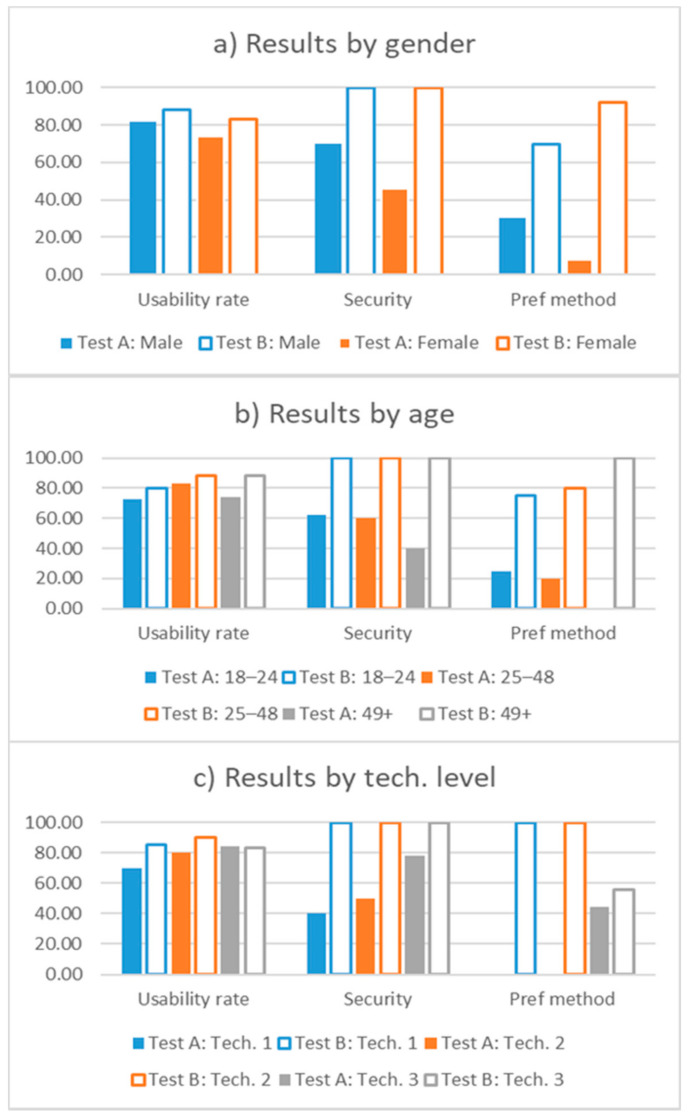
Usability study detailed results.

**Table 1 sensors-21-05716-t001:** Exposure of methods A and B to attacks defined in [13]. Attacks in the first, second, and third groups are written in yellow, green, and red, respectively.

Attack Vector	A	B
4.1. Insufficient Redirect URI Validation	Yes	Yes *
4.2. Credential Leakage via Referer Headers	Yes	No
4.3. Credential Leakage via Browser History	Yes	No
4.4. Mix-Up Attacks	Yes	Yes *
4.5. Authorization Code Injection	Yes	No
4.6. Access Token Injection	Yes	No
4.7. Cross Site Request Forgery	Yes	No
4.8. Access Token Leakage at the Resource Server	Yes	Yes
4.9. Open Redirection	Yes	No
4.10. 307 Redirect	Yes	No
4.11. TLS Terminating Reverse Proxies	Yes	Yes
4.12. Refresh Token Protection	Yes	Yes
4.13. Client Impersonating Resource Owner	Yes	Yes
4.14. Clickjacking	Yes	No

*: The attack is significantly harder to develop using the method B compared with the method A.

**Table 2 sensors-21-05716-t002:** Participants’ demographics.

Individual Characteristics	N	%
**Gender**		
Male	10	43
Female	13	57
**Age**		
0–24	8	35
25–48	10	43
49+	5	22
**Tech. Level**		
Occasional user	10	43
Habitual user	4	17
Advanced user	9	39

**Table 3 sensors-21-05716-t003:** Usability study overall results.

Question	Test A	Test B
Usability	7.74	8.52
Security	13	23
Pref. method	4	19

**Table 4 sensors-21-05716-t004:** Participants’ commentaries.

Participant Comments	#
**Test A**	
It is more familiar	3
The interaction is simpler	2
Seeing the link gives me more security	1
**Test B**	
The interaction is simpler	6
It is more secure	14
Do not have to leave the app	15
I do not feel comfortable clicking a link	16

## Data Availability

The data presented in this study are available on request from the corresponding author. The data are not publicly available due to privacy reasons.

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
