# Peer review of "Authorizing Third-Party Applications Served through Messaging Platforms"

_sensors, 2021, doi:10.3390/s21175716_

Round 1

Reviewer 1 Report

This article addresses a timely need of authorizing third-party applications served through messaging platforms, making sure that no MitM risk exists without requiring to use additional medium, e.g. an external web browser, and then return to the platform. The papers organization helps the reader to go over an adequate literature review, covers most of the needed concepts (control background) and details the proposed grant type and security analysis with a usability study. I have only 2 major points to question, 1) new updates on apps like telegram allow for embedded web-browser experience, this has to be taken into account in the usability (contrasting with Signal and Whatsapp), and 2) the number of participants seems to be rather small to derive a good confidence in the observed results. The reviewer recognizes that both observations can be quite impactful to the results presented in the paper and suggests addressing them with adequate reflection on the conclusion and pertinent places in the text (especially the Telegram feature of embedded web browser).

Reviewer 2 Report

The paper is well written and supports the currently significant problem. However, some parts of the manuscript may be enriched. Firstly, an in-depth literature review should be arranged. It would be necessary to ground the presented study in the existing literature, present the problems that researchers have considered in recent times. Moreover, it should be mentioned how other scholars deal with described attacks.

What are the paper’s limitations? Future research? These are not specified in a clear way and should be included in the concluding section.

Reviewer 3 Report

The authors present a very interesting problem related to the security of authorization, on the example of authorization in messaging such as Whatsup, Viber, Signal etc. The authors focus on the ManInTheMiddle attack. The article describes in detail the problem with the correctly identified problem. Although the structure of the article is not typical of IMRAD, the structure of the article is favorable to the reader. However, I have doubts, without questioning the authors' achievements, whether the length of an article is sufficient as an "article" and whether it should not be "communitacation".
Undoubtedly, an interesting test of the authors' idea is the usability study, in which the authors present the study of their idea on a group of users. The study itself is well designed and described. However, I have to point out that it would be better to enlarge the group of tested users. The obtained results would certainly be more reliable.
In Table 3, the authors use the precision to two decimal places in the first line. The rest should be the same. Additionally, decide whether it should be a comma or a dot.
Interesting article.

Reviewer 4 Report

This paper proposes a new type of OAuth grant type for the interaction between relying party (i.e., resource owner) and client via messaging platforms. The paper specifically aimed at preventing the Man-in-the-Middle attack and improving the user experience.

In the first part of the research, the authors described their protocol and its security analysis. Authors should clarify how it differs from similar work in the literature rather than just comparing to the original OAuth protocol (more review of related work should be added e.g., how the others improve the MitM issues in OAuth). The analysis from Line 333 to 336 is oversimplified, in which the authors claimed that the protocol could prevent eight types of attacks simply because they are “tightly coupled to the redirection mechanisms” (what is tightly coupled? why can’t the other attacks be considered tightly coupled?). Such analysis does not give the protocol any sense of stronger security. If the authors cannot provide a proper analysis for these attacks, they should exclude them from the section. The authors should also clarify that their Security Analysis section focuses only on the MitM attackers and compares the capability to counter the MitM attacks with that of the original OAuth protocol. This is not the same as normal security analysis, in which more threat models can be involved (e.g., DoS, eavesdropping) and the protocol should also be considered from different security aspects (e.g., is there any impact of using authentication server to ask for end-user consent? I suspect that there will be issues of server overloading or increasing the chance of DoS attack on the server).

In the second part, the authors evaluated the usability of the proposed protocol by surveying/interviewing people using two versions of their authentication app. However, the factors influencing the usability were not specified. The apps were poorly designed as there wasn’t much of differences between them. I can hardly tell whether the new design can improve any of the common usability factors (e.g., the 5E in the literature Effectiveness, Efficiency, Engagement, Error Tolerance, Ease of Learning). The results of the survey also raise a question of user acceptance on the proposed solution as users may not trust the app if they do not know what happens behind the authentication bot. It would also be interesting to see how users think if the link in app A cannot be examined at all (e.g., using short form such as bit.ly link).

The authors should also revise and improve their writing. For example, some sentences are very long which makes them difficult to understand (e.g., Line 234-239). All the “section”, “fig” should be capitalised when referring to specific sections or figures. Fig 2a and 2b captions should be Figure 5 and 6. Line 237: “as that which was stored” -> “as that stored”; Line 513 “more highly” -> “higher”, etc.

Round 2

Reviewer 4 Report

I agree with most of the authors' responses, apart from Figure 3 that needs a bit more clarification. From the figure, it seems that the main difference of the proposed protocol from the original is that the authentication server asks the end user to give consent (apparently that is why these steps are coloured differently in orange). However, the authors stated in their response that the OAuth already works this way. Therefore, the authors should highlight the differences between the two protocols in greater detail in this figure. It is also difficult to compare between Figure 1 and Figure 3 – why the arrows are drawn differently.
